# Influence of Filler Metal on Electrochemical Characteristics of a Laser-Welded CoCrMoW Alloy Used in Prosthodontics

**DOI:** 10.3390/ma15165721

**Published:** 2022-08-19

**Authors:** Lukasz Reimann, Zbigniew Brytan, Grzegorz Jania

**Affiliations:** 1Materials Research Laboratory, Faculty of Mechanical Engineering, Silesian University of Technology, Konarskiego St. 18a, 44-100 Gliwice, Poland; 2Department of Engineering Materials and Biomaterials, Faculty of Mechanical Engineering, Silesian, University of Technology, 44-100 Gliwice, Poland; 3Dental Engineering Laboratory Grzegorz Jania, 48-200 Prudnik, Poland

**Keywords:** CoCrMoW alloys, CoCr alloys, stainless steel, LBW, corrosion resistance, EIS, dentures, prosthodontic, biomaterials

## Abstract

This paper sought to determine corrosion resistance changes in the artificial saliva of a CoCrMoW-based alloy used for dental prostheses under Nd:YAG laser welding with CoCr alloy and stainless steel wire filler metals. The paper presents the corrosion characteristics of such joints, including the next stage of porcelain-fused-to-metal (PFM) firing. Corrosion tests were performed by electrochemical methods registering anodic polarization curves and electrochemical impedance spectroscopy (EIS). The microstructures were assessed by scanning microscopy (SEM) and chemical composition analysis (EDS) at the connection and heat-affected zones. Welding CoCrMoW alloy with and without a filler material increased the open circuit potential of the samples by 40–100 mV compared to unwelded base alloy. At the same time, a potentiodynamic test showed a polarization resistance R_pol_ reduction in welded samples, both for CoCr and stainless steel wires, as compared to the base CoCrMoW material. On the other hand, when comparing the current density and polarization resistance between materials welded with two different filler metals, better results were obtained for samples welded with stainless steel wire. The polarization resistance R_pol_ for the base alloy was 402 kΩ·cm^2^, for the CoCr wire weld it was 436 kΩ·cm^2^, and the value was 452 kΩ·cm^2^ for stainless steel wire welds. Comparing polarization resistance R_pol_ from the Tafel analysis and the total charge transfer resistance from Rp_(EIS)_ from EIS, the CoCrMoW alloy welded with a stainless steel wire after heat treatment equaled or even slightly exceeded the corrosion resistance of the base alloy and alloy welded with dedicated CoCr wire after heat treatment. These results indicated the possibility of using stainless steel wire for the laser welding of CoCrMoW alloys dental prostheses, including the next stage of PFM, without sacrificing the corrosion resistance of such connections, and this was confirmed by most electrochemical parameters.

## 1. Introduction

Welding technology has been known in prosthodontia for many years. Most prosthetics laboratories use it successfully with nonprecious alloys (titanium alloys, cobalt alloys, and nickel alloys) and noble metal alloys. That method repairs broken fixed dentures and attachment dentures, and connects long prosthetics (bridges) and dental implants—fabricated in part because it minimizes potential distortion problems that may occur after welding, relative to casting technology [1,2].

There are many joining technologies employed in prosthodontia for preparing dentures, including soldering, TIG (tungsten inert gas) welding with or without a filler metal (autogenous TIG), and laser beam welding (LBW). Presently, the most promising welding method in dental engineering is LBW because it does not require a unique filler material; this prevents a galvanic cell between materials and a subsequent reduction in corrosion resistance [3]. However, suppose that a discontinuity (resulting, for instance, from the preparation of two parts of one prosthesis to be joined, which may be 0.5–1.0 mm) in a prosthetic construction is more significant and requires a filler material made from alloys with chemical compositions most closely aligned to the welded material. As in the first case, when no filler material is used, the second method with filler material does not prohibit stress formation in the weld or grain growth in the structure, which deteriorates the mechanical properties of welded elements and reduces corrosion resistance.

Several studies have cited many welding examples of repairing or constructing advanced long dentures that focused primarily on titanium alloys, and the influence of laser processing conditions on the joint properties [4,5,6,7]. Cobalt–chromium (CoCr) alloys are trendy metallic materials intended to produce dentures. However, relatively few of the literature reports cite the use of laser welding on these alloys in prosthetics construction, and most of that work focused solely on the mechanical properties of the joints [1,8].

These CoCr-based alloys possess high biocompatibility, excellent corrosion resistance, and compatibility with dental ceramics as aesthetic components. Initially, CoCr alloys were alloyed with Mo and provided good casting material fluidity. They were gradually combined with W to increase the anticorrosion behavior after additional alloy casting. The CoCr-based alloys with W demonstrated higher electrochemical stability in artificial saliva compared with CoCr-based alloys without W [8]. Furthermore, Mo, similarly to Cr, improves resistance to localized corrosion attacks in chloride-containing environments, such as bodily fluids. The Cr primarily enhances corrosion resistance, while Mo affects grain refinement and matrix strengthening. The tungsten decreases the coefficient of thermal expansion of the alloy, and CoCrMoW alloys result in a thinner oxide layer formation as compared to CoCrMo alloys. This improved the strength of the metal/ceramic bond during porcelain fusion to metal (PFM), which reduces the risk of porcelain cracking [9,10].

The CoCrMoW-based alloy microstructure in the as-cast state consists of a dendritic γ-fcc (face-centered cubic) metastable matrix and inter-dendritic segregation of intermetallic compounds (carbides, intermetallic phases, such as the the σ phase, and complex lamellar structures). The γ-fcc may undergo a martensitic transformation to ε-hcp (hexagonal close-packed) during heat treatment, combining solution annealing and isothermal aging [11,12]. Carbide precipitation along the grain boundaries and inter-dendritic regions strengthens the as-cast condition. During subsequent welding, e.g., repairing dentures, the microstructure undergoes melting and solidification, which affects the morphology and composition of the newly formed microstructure. More intense segregation of alloying elements and precipitation of secondary phases occurred due to the high cooling rate (usually brittle, lowering plasticity). As a result, cracking may occur in the welded zone [13]. Therefore, selecting the appropriate welding parameters is essential for repair success. Moreover, welding dissimilar materials, including stainless steel, affects weld zone properties.

One of the criteria in the selection of denture materials is corrosion resistance, and some authors [14] reported that welded joints had lower corrosion resistance than the base material. The selection of filler materials, presented in this study, was made according to their chemical composition that should be as close as possible to the chemical composition of the welded alloy. Therefore, one of the materials is chromium–cobalt wire with molybdenum addition, and the other is stainless steel with a chromium content similar to that of the base alloy. The sparse literature data on the corrosion behavior of welded CoCrMoW dental alloys in different configurations (such as stainless steel and a dedicated cobalt alloy wire) guided this work. Therefore, this study evaluated the corrosion resistance of LBW joints in various configurations of a CoCrMoW alloy welded with nickel-free stainless steel cobalt–chromium alloy wire, as well as autogenous welding in an artificial saliva environment after an additional heating stage from 920–950 °C, simulating the porcelain-fused-to-metal (PFM) firing process.

## 2. Materials and Methods

This research was performed on the cobalt–chromium–molybdenum–tungsten (CoCrMoW) alloy called Remanium 2000+ (Table 1), which is used as a biomaterial in prosthodontics. The samples were cast in a centrifugal casting machine, namely the Fornax T (Bego, Germany), according to denture manufacturing procedures [15]. The sample dimensions were 10 × 10 mm by 1 mm thick. Noting the fact that CoCrMoW alloy sees use in complex dentures with ceramic buildups, a set of samples was heat treated after casting (denoted as HT in the sample name), which corresponded to porcelain-fused-to-metal (PFM) firing (Table 2), using a VKM Master unit made by the Vita company.

Next, the samples were laser welded using a laser welding machine (Nd:YAG LaserStar Plus, Bego, Germany) with the following parameters: 8 ms pulse duration, spot diameter 0.7 mm, and a voltage between 270–290 V. Three sample sets were welded, two with filler materials (different) and one without filler material. The filler materials were the nickel-free stainless steel wire Menzanium^®^ (Scheu Dental, Iserlohn, Germany, denoted as ST in the sample name) and cobalt–chromium wire (Schütz Dental GmbH, Rosbach, Germany, indicated as CO in the sample name). Table 1 lists their chemical compositions. After welding, all samples were ground with SiC paper, grades 500, 800, and 1200, and polished with a diamond suspension of 3 µm. The samples were marked in the following way: Base—base CoCrMoW alloy, HT—heat treatment, 0—laser welding without filler metal (autogenous welding), ST—welded with stainless steel wire, and CO—welded with CoCr alloy wire.

Two methods examined the electrochemical properties and characteristics of these materials, namely potentiodynamic polarization and electrochemical impedance spectroscopy. Corrosion tests were performed on an Atlas 0531EU & IA (Atlas-Sollich, Rebiechowo, Poland) potentiostat station in a water environment simulating artificial saliva prepared based on the Fusayamya–Meyer formula (Table 3) at room temperature (23 °C). Electrochemical tests were carried out in three-electrode corrosion cell systems according to the PN-ISO 17475:2010 standard, with the test sample as the working electrode, an Ag/AgCl reference electrode (potential 207 mV at 25 °C), and a platinum wire as the auxiliary electrode. Polarization tests were divided into the following two stages:Determination of the open circuit potential (E_ocp_) in electroless conditions over 1 h;Anodic polarization recording potential variation from E_ocp_-100 mV with step and potential speed changes of 1 mV/s until reaching a current density of 1 mA/cm^2^. The polarity was reversed, and the curve to the initial potential was recorded.

**Table 3 materials-15-05721-t003:** Composition of artificial saliva according to the Fusayamya–Meyer formula [16].

Element	NaCl	KCl	NaH_2_PO_4_·2H_2_O	CaCl_2_·2H_2_O	Na_2_S·9H_2_O	Urea	Distilled Water
Content	0.4 g/L	0.4 g/L	0.69 g/L	0.79 g/L	0.005 g/L	1.0 g/L	1 L

Tafel extrapolation using the AtlasLab software established the characteristic parameters related to electrochemical corrosion, which included current density (J_cor_), corrosion potential (E_cor_), and polarization resistance (R_pol_), which was determined according to the Stern–Geary Equation (1), where b_a_ and b_c_ are the slope of the anode and cathode sections of Tafel, respectively. The mass corrosion rate of the tested materials was also determined using Equation (2), where M is the atomic mass of the metal, z is the number of electrons exchanged in the anode reaction, F is the Faraday constant, and J_cor_ is the corrosion current density. The breakdown potential (E_br_) was determined as a place of depassivation and the inflection of the anode curve, as well as the method of determining the value of repassivation potential (E_rp_) as the point of intersection of the return and primary curve.
(1)$$R_{pol}=\frac{b_{a}\cdot b_{c}}{2,3\cdot i_{cor}\cdot \left(b_{a}+b_{c}\right)}$$
(2)$$r_{cor}=\frac{M}{zF}J_{cor}$$

Electrochemical properties were also determined using the second method, namely electrochemical impedance spectroscopy (EIS), first by stabilizing the samples in the test environment for 15 min without current flow, and then with a flow through the solidified AC system at an amplitude of 10 mV at frequencies from 100 kHz to 10 mHz. The results are presented as Nyquist and Bode plots. An electrical equivalent circuit (EEC) was assigned to reproduce the relationships appearing in these studies using the AtlasLab and EC Lab software, in which the numerically generated curves were fitted to the experimental results. Apart from typical resistors, non-linear CPEs (constant phase elements) were adopted in the EEC.

Two tests per sample were performed in the EIS and potentiodynamic tests. The obtained values were very close to each other, thus, one representative result was selected and presented in the article.

After corrosion resistance tests, a light microscope (Axio Observer-Zeiss) and a scanning electron microscope (SEM, Supra35-Carl Zeiss AG) examined the microstructures and surface degradation of the samples. Chemical compositions in microregions were performed via EDS spectroscopy during SEM observations (acceleration voltage, 10 kV).

## 3. Results

### 3.1. Potentiodynamic Polarization Curves

To determine the corrosion resistance of laser beam-welded CoCrMoW alloy in different filler material configurations and heat treatments, the potentiodynamic tests preceded by registration of steady-state potential in a water environment simulating an artificial saliva solution (Figure 1a), next to the polarization curve with reverse anodic scan, were registered (Figure 1b,c). Table 4 shows the Tafel analysis results.

Open circuit potential (E_ocp_) of tested materials varied from −277–(−178 mV), and higher levels were recorded for laser welded samples with filler materials. The registered upward trend, observed in Figure 1a in the study of the stationary potential, may be related to the appearance of its ions at the metal–electrolyte interface in order to create an equilibrium state. The heat treatment influence, which simulated porcelain-fused-to-metal (PFM) firing, was seen in the E_ocp_ values. The base alloy increased from −277 to −198 mV; the sample welded with stainless steel wire decreased from −178 to −215 mV. In two other cases (autogenous welded CoCrMoW and welding with CoCr filler materials), there were no significant differences in the E_ocp_ values. The same relationship was observed when comparing corrosion potential (E_cor_) values.

The tested materials were polarized at a rate of 1 mV/s until a current density of 1 mA/cm^2^ was reached, and then the direction of polarization was reversed. The determined characteristic potentials allowed us to distinguish the passivity zone, visible as a section of the horizontal line after the corrosion potential peak in Figure 1c,d. For potentials higher than E_br_, the pitting initiation can be observed, while for potentials more negative than E_rp_, the existing pits were repassivated. A path in the range of the E_br_ to E_pr_ potentials was safe with regard to new pits, but those that exist on the surface could develop.

The corrosion current density (J_cor_) after heat treatment of the weld configurations showed no significant changes, and the biggest difference was recorded when comparing the base alloy without heat treatment and after the PFM treatment, as the corrosion current increased from 21 to 41 nA/cm^2^. In all other cases, the differences were <14 nA/cm^2^.

Analysis of potentiodynamic curves (Figure 1b–d) determined two characteristic electrochemical parameters, namely breakdown potential (E_br_) and repassivation potential (E_rp_), which describes the corrosion resistance of materials. The first represents when pitting appears on the surface, and the repassivation potential stipulates when new corrosion damage does not occur. Both potentials were close when assessing heat treatment and welding with or without filler material. The loop formed in the diagram (Figure 1c,d) was very narrow, E_br_ 920 ± 12 mV and E_rp_ 932 ± 31 mV on average.

Comparing the polarization resistance (R_pol_), Table 4 shows that the polarization resistance obviously decreased for the CoCrMoW alloy welded with both filler materials (cobalt, or CO, and stainless steel, or ST, wire)—from 634 kΩ·cm^2^ to 400 and 450 kΩ·cm^2^, respectively. Notably, a lower decrease occurred for stainless steel than for the CoCr alloy. Subsequent heat treatment (simulation of the PFM firing process) for welds with filler materials did not result in significant polarization resistance changes. The R_pol_ for the base alloy was 402 kΩ·cm^2^, 436 kΩ·cm^2^ for CoCr wire welds, and 452 kΩ·cm^2^ for stainless steel wire welds. An interesting phenomenon was observed for the autogenous welded alloy (Base_0), where the heat treatment significantly increased the polarization resistance from ~550–800 kΩ·cm^2^.

The mass corrosion rate (r_cor_) relates directly to the corrosion current density, and it ranged from 0.2–0.6 g/m^−2^·year^−1^ (Table 4). The CoCrMoW alloy after heat treatment (Base_HT) had the highest corrosion rate (0.58 g/m^−2^·year^−1^). On the other hand, all welded joints with various filler materials had lower corrosion rates. For autogenous welds and welds made with CoCr wire, the corrosion rates were slightly lower (~0.50 g/m^−2^·year^−1^), while for stainless steel welds, it was 0.41 g/m^−2^·year^−1^, lower than the base CoCrMoW alloy. There was a noticeable relationship between heat treatment and the corrosion rate, which decreased for heat-treated autogenous and CoCr wire welds. However, heat treatment increased the corrosion of stainless steel welds.

### 3.2. Electrochemical Impedance Spectroscopy

Electrochemical impedance spectroscopy (EIS) tests were performed in artificial saliva to characterize the electrical properties and to analyze the interfacial properties of welded joint configurations. The EIS results are presented as Nyquist (Figure 2a) and Bode (Figure 2b,c) diagrams. The Bode plot in Figure 2b separately presents the impedance modulus to amplitude relationship, while Figure 2c shows the phase angle dependency. The changes in the impedance modulus of the corrosion systems allows for the supposition that its higher value is related to the better barrier properties of the surface and, thus, that it has better corrosion resistance.

By analyzing the course of the impedance spectra and the value of their slope coefficient on the Nyquist diagram (shown in Figure 2a), it can be seen that, in the entire studied range, the CoCrMoW base alloy (Base) and the base alloy with subsequent heat treatment without filler metal (Base_HT_0) were characterized by significantly lower susceptibility and speed to corrosion damage. This was deduced from the high impedance (imaginary part of impedance ImZx) at the variable frequencies for those samples. Identical results were recorded for high-frequency values (the left side of the X-axis on the Nyquist plot, magnified), which showed the highest curve slopes and corresponded to the lowest corrosion rates.

Comparing the shapes of the impedance spectrum curves of samples welded with the filler materials, two frequency ranges were observed, once with cobalt wire welds and another time when the stainless steel wire welds showed better results. From 100 kHz to 660 mHz, the cobalt wire welds offered greater corrosion than stainless steel welds. However, at very low frequencies (100 kHz to 660 mHz), stainless steel wire had better resistance (Figure 2a).

The Bode impedance curves shown in Figure 2b tracked the corrosion system behaviors and showed a characteristic impedance decrease with an increasing electric signal frequency.

The highest impedance values in the frequency range shown occurred for base material (Base) and autogenous welded material (Base_0) samples. The lowest impedance over almost the entire frequency range (~0.5 Hz to the high frequencies), was found for the cobalt wire (Base_CO sample). Comparing the impedance and phase angle results for the samples welded with the filler materials (see Figure 3), the most significant differences occurred in the narrow frequency ranges. Significant differences in the case of the impedance spectrum occurred from 50 Hz to 1000 Hz, as presented in Figure 3a.

The Bode diagram plotted in Figure 2c shows the phase shift angle (φ) dependence on the impedance modulus and allows for the assessment of how well Cl^−^ ions transmit across the joint surfaces and contribute to corrosion degradation. The value of the phase shift angle, i.e., the delay of the system’s response to the voltage signal, changes the impedance value. In the case where the current and voltage are in phase, the impedance can be expressed as pure resistance; when φ takes values close to −90° the impedance can be described by a capacitor, and when φ is a value close to +90°, it can be described as an inductor. The course of the phase shift angle curves and its value allow the assessment of the quality of the protective barrier against corrosion damage on the surface of the tested material. The smallest value of the phase shift angle allows us to assume that this barrier is the thickest and most compact.

All materials yielded a high phase shift angle (~75°) covering almost 25% of the tested frequency range; the highest values were obtained for the base material (Base) and the base material after heat treatment welded without a filler metal (Base_HT_0).

In Figure 2c, in the high frequency range of 40–100 kHz, a decrease in the value of the phase angle was observed, corresponding to the assumption of positive values by the impedance component ImZx. This phenomenon is probably related to the fact that, in high frequencies the impedance of the parasitic capacitor decreases while the impedance of the inductor increases, and the modeled element behaves similarly to an inductor.

By comparing the results of the CoCrMoW samples welded with filler materials (Figure 3b) in the narrow frequency range (0.1–100 Hz), higher phase shift angles (~5°) occurred for CoCr wire welds in samples with and without heat treatment. The EIS demonstrated higher impedance modulus for stainless steel wire, over essentially the entire frequency range, though samples welded with CoCr wire had a more favorable phase shift angle by 3–5°.

For the impedance curves, an equivalent electrical circuit (EEC) was fitted that best described the corrosion system (Figure 4). The EEC consisted of two R-C groups disposed in series, two constant phase elements CPE, and three resistors. The resultant EEC impedance was expressed using Equation (3). The EIS results are a good fit with the EEC proposed.
Z = Rs + 1/(1/R1 + (Y_1_ (jω))^n1^) + 1/(1/R2 + (Y_2_ (jω))^n2^)(3)

The physical meaning of the EEC proposed [17] is attributed to the electrolyte resistance (Rs), i.e., in Fusayamya–Meyer solution, R1 to the resistance oxide, the resistance of the pores in the passive film formed at the surface, the pore capacitance of the passive film, which was expressed here by the constant phase element (CPE_1_), R2 is the charge transfer resistance and the capacitance of the electric double layer at the interface of the metallic material (alloy) expressed by the constant phase element (CPE_2_). The EEC model was very similar to that from Uriciuc, W.A. et al. [17] for their study of nickel- and cobalt-based dental alloys in ringer solution, as well as the work of M. Meticos-Hukovic et al. [18] for CoCr alloys dipped in Hank’s solution. The proper elements were assigned to the electrical equivalent circuit, and it was found that these values enabled the assessment of their variability and characterized the corrosion processes that occurred on the surface (Table 5). The constant phase element (CPE) included in the EEC was characterized by a constant phase shift angle. The following expression describes the impedance of the CPE: Z_CPE_ = 1/Y^0^(jω)^n^, where Y_0_ and n are parameters related to the phase angle. The more heterogeneous corrosion processes on the metal surface, the lower the n value becomes. When *n* = 1, this implies an ordinary capacitor, but a resistor is implied if *n* = 0, and *n* = 0.5 corresponds to the Warburg element used to model diffusion. The CPE1 elements (expressed by Y_1_ and n_1_) in this system behaved similarly to a capacitor, while CPE_2_ (represented by Y_2_ and n_2_) had variable resistor/capacitor characteristics depending on the sample. Comparisons of the corrosion resistance in a given system are made based on the resistance R2, related to the charge transfer resistance and the capacitance of the electric double layer at the interface of the metallic material (the most significant differences for samples welded with cobalt and stainless steel wire). The R1 related to the resistance of the pores in the oxide film varied from the lower value (0.01 kΩ) for the unwelded sample and (0.1 kΩ) for the autogenous welded base alloy to (1–2 kΩ) in most cases to (14 kΩ) for the stainless steel welded joint. When considering the practical use in dental prosthetics, the most critical stage of material processing occurs after the heat treatment that simulates PFM. In this context, the highest charge transfer resistance R2 (127–204 kΩ) is shown in the autogenous welded material (CoCrMoW alloy welded without the filler metal), and then in materials welded with the filler materials. The R2 values were remarkably similar for the CoCr wire welds (83–90 kΩ) and the stainless steel wire welds (89–98 kΩ). Interestingly, after the stainless steel welds, the corrosion resistance eclipsed the CoCr wire welds.

### 3.3. Microstructural Characterization

Figure 5 compares the microstructures of welded material configurations in as-welded conditions and after heat treatment (HT), simulating (PFM) firing. The typical as-cast dendritic microstructure with inter-dendritic lamellar microstructures of secondary precipitates is shown for CoCrMoW in Figure 5, base, where the interdimeric regions became thicker and more visible after heat treatment (Figure 5, Base_HT). The pitting corrosion preferentially occurred in the inter-dendritic regions. The use of filler materials during welding resulted in structure refinement compared to the non-welded material, which was associated with an extremely fast cooling rate after LBW. Similarly, in the laser-welded zone of the parent material as seen in Figure 5, base_0, and base_HT_0, a dendritic microstructure occurred, and the heat-affected zone of transient microstructure was readily identified. The corrosion degradation on the surface proceeded into the weld area. Comparing the microstructure of samples welded with CoCr (Figure 5, Base_CO, Base_HT_CO) wire to samples welded with the stainless steel wire shown in Figure 5, (Base_ST, Base_HT_ST), the filler material did not involve significant differences in surface corrosion damages caused by testing. On the other hand, when observing the CoCrMoW substrate alloy, we concluded that greater corrosion damage occurred in the case of welding with stainless steel wire (Figure 5, Base_ST, Base_HT_ST).

Moreover, SEM observations and EDS analyses were performed after corrosion tests on samples welded with filler materials, and these were presented in Figure 6 and Figure 7. Three different areas were revealed, namely the base material (area 1), the transient zone, the heat-affected zone (HAZ area 2), and the weld metal zone (area 3). The chemical compositions of welded samples were compared before and after heat treating between 920–950 °C, simulating porcelain-fused-to-metal (PFM) firing.

The chemical composition of the CoCrMoW alloy welded with cobalt wire (Figure 6, Base_CO) in the base cast alloy (area 1) and the transient zone (area 2) did not change significantly and retained the base CoCrMoW alloy composition, though clear differences in the weld metal (area 3) were observed. This area was enriched in Mn, slightly enriched in Cr, and depleted in W, due to the Co wire’s metal composition. The weld metal (area 3) was diluted by ~50% with the base material. The EDS microanalysis revealed that the transient zone (area 2) between the CoCrMoW cast alloy and cobalt welding wire was enriched in Mo, Cr, and Si from the cast alloy, which was concentrated predominantly in inter-dendritic regions. At the same time, cobalt remained in the solid solution dendritic regions. The fast and unstable solidification from welding temperature resulted in chemical segregation. The phase that precipitated during the solidification of the CoCrMoW alloys lacked the alloying element. Thus, decreasing chromium and molybdenum concentrations enriched the inter-dendritic spaces in these elements. At the inter-dendritic regions, various secondary phases formed; depending on the amount of carbon in the alloy and other alloying elements, precipitation continued due to a eutectic reaction or simple precipitation. Here, M_23_C_6_ carbides dominate in CoCrMoW alloys due to high levels of Cr and Mo, but also fewer complex carbides, such as M_7_C_3_, M_2_C_2_, M_6_C, and MC, and many intermetallic phases, such as the σ phase [19]. Inter-dendritic regions were mostly uniform, without complex lamellar structures composed of σ phase and carbides. Moreover, the concentration of carbon in the alloy was limited; thus, more probably, the inter-dendritic regions comprised intermetallics rich in Co, Cr, and Si, such as the σ phase, χ phase, Co_5_Cr_3_Si_2_, and M_23_C_6_ carbides [12,19]. Future efforts will focus on detailed analyses of the inter-dendritic zone microstructures and precipitates.

Soaking at 920–950 °C (PFM firing process) of the welded material affected the homogenization of the transition zone (area 2) in terms of chemical composition that leaned towards a composition similar to the CoCrMoW base alloy shown in Figure 6, Base_HT_CO. The Mo levels decreased while the W levels increased. Moreover, the microstructure became more cellular-dendritic with well-defined individual cells. A similar relationship with higher Cr levels was observed in the weld metal (zone 3) for a heat-treated sample, while Mn levels decreased to ~3% from 15% in the cobalt welding wire.

When analyzing CoCrMoW cast alloy welded to stainless steel, as presented in Figure 7, Base_ST, a transient zone (zone 2) between cast alloy and weld metal was also saturated in iron as nitrogen from the stainless steel wire. Manganese diffused weakly into this area. The weld metal (zone 3) was diluted at ~30% with the base material, showing c.a. 18% chromium, 12% cobalt, 10% manganese, and 1.5% tungsten. The soaking at 920–950 °C affected the interdiffusion process between materials (Figure 7, Base_HT_ST), which enriched the base CoCrMoW cast alloy in chromium. The transient zone (zone 2) was still enriched in iron and depleted in manganese. This zone also grew due to diffusion, which showed a cellular microstructure with a strong segregation of alloying elements on coarsening cellular grain borders.

## 4. Discussion

The application of laser welding technology to materials and products intended for dental engineering is promising and has seen significant attention recently. This is due to LBW, because it reduces the HAZ, and the weld provides similar strength properties to the welded material. This maintains the geometric features necessary for the correct fit to the anatomy of the tissues in the oral cavity. In addition, it suitably repairs cracks and joins segments of extensive prosthetic works, whose execution in the form of castings may deform, as well as due to the low thickness of joint welding in the region near porcelain and resins. Furthermore, corrosion resistance research of variously prepared joints [20] showed a significant decrease in released ions (>100) from each sample for LBW joints when using a Nd:YAG laser compared to the traditional soldering joint. This aspect was essential due to the risk of heavy metal absorption present in the prosthetic alloys (Cr, Co, Mo, W, etc.) via oral mucosa and other routes. Similarly, favorable results apply to the mechanical properties of LBW joints. In this respect, comparing the fracture strength for joints achieved with different methods (soldering, TIG, LBW) with and without filler material [21] revealed better results for laser-welded materials with the use of the filler metal.

When comparing the steady-state potential under currentless conditions, it was found that autogenous welding and welding with the filler materials increased potential value (shifted to more positive potentials) relative to the non-welded base alloy. The E_OCP_ increased by 41 mV for sample Base_O, 88 mV for sample w Base_CO, and 99 mV for sample Base_ST. A similar relationship was previously reported in [22], where remelting using a Nd:YAG laser increased potentials by 50 mV for the Ag–Pd–Au–Cu substrate alloy, which suggested [23] a relationship to elevated levels of elements that positively influenced corrosion resistance. The influence of alloying elements on the corrosion resistance of Co-Cr alloys used in dental prostheses was reported [24], and increased chromium levels increased the E_OCP_ potential. The EDS analysis for samples after welding using filler materials, as presented in Figure 6 and Figure 7, confirmed increases in chromium, molybdenum, and manganese in the HAZ areas and the welded joint areas, as compared to the base material. This indicated an active relationship between these alloying elements and the E_OCP_ value.

The potentiodynamic curve analyses and the Tafel extrapolation results did not reveal any significant differences in the values of the parameters, suggesting deterioration of the corrosion resistance concerning the base material. They even indicated a slight improvement in the corrosion potential and polarization resistance. On the other hand, the opposite behavior for materials subjected to heat treatment (simulating the firing of porcelain on a metal substrate, known as PFM) was observed, for which the electrochemical parameters decreased slightly. Changes in electrochemical properties related to corrosion resistance for the CoCrMoW base alloy were reported previously, for example in [25], in which the material had a slightly lower corrosion resistance after heat treatment than the base material after casting, as evidenced by ~50% lower polarization resistance and higher corrosion rate (lower slope angle of the impedance spectrum curve in the Nyquist diagram). The decrease in corrosion resistance and lower Rp after simulated heat treatment (PFM) may result from a structural change after firing as previously reported [26].

Slight differences were also observed after EIS tests in the alternating current system. For nearly the entire frequency range (from 100 kHz to 0.66 Hz), the impedance spectra presented in the Nyquist diagram plotted in Figure 2a were very similar, and the a1 coefficient values were also similar (0.87–0.91), as were the characteristics of the constant-phase element CPE1 (see Table 5) and the inclination angle of curves to the ordinate axis shown in Figure 2a.

Evaluating the results of electrochemical corrosion resistance tests (Table 4) for samples welded with cobalt or stainless steel wire concerning the base material, a slight reduction in corrosion resistance in the environment of artificial saliva was observed. A decrease in the polarization resistance (R_pol_) by ~200 kΩ·cm^2^ and ~100–150 kΩ·cm^2^ relative to the autogenous weld (without filler material) of the CoCrMoW base alloy confirmed this. Similarly, a slight reduction in the corrosion resistance of LBW stainless steel was reported [26], and a TIG welded joint showed lower/negligent anti-corrosion properties [27].

Comparing the Tafel analysis results with the impedance spectroscopy results, the corrosion resistance of samples welded with filler materials in respect to unwelded and autogenous welded deteriorated, as evidenced by several factors. First, there was a significantly lower charge transfer resistance, R2, that decreased from 198 to 80–90 kΩ. Second, there was a narrower phase shift angle frequency range, reaching a maximum of ~75°, which relates to the lower quality and consistency of the surface layer. Third, the flattening of semi-circles in the low and very low frequency range, visible in the Nyquist diagram (Figure 2a) was related to chemical composition differences and inhomogeneities on the alloy surface and the filler material used for welding. Considering the above statements, analyzing the data from the EIS tests for samples welded with the filler materials, and the characteristics of the Bode diagram plotted in Figure 3b, the lower phase shift angles and the narrower peak range for the stainless steel wire-welded sample were due to the greater diversity of its chemical composition than the cobalt-welded sample, relative to the base material.

A summary of the electrochemical tests was made based on resistance R_pol_ from the Tafel analysis and the total charge transfer resistance of the equivalent system from EIS measurements, where the resistance is summarized as Rp_(EIS)_ = R1 + R2 (the oxide layer resistance, the resistance of the pores in the oxide layer formed at the surface, and the charge transfer resistance of the electric double layer at the interface of the metallic material), as shown in Figure 8. In such a combination, the CoCrMoW base alloy result after heat treatment was taken as the baseline corrosion resistance (dashed lines indicate the level of polarization resistance obtained from the two methods). The CoCrMoW alloy welded with a stainless steel wire after heat treatment equaled or even slightly exceeded the corrosion resistance of the CoCrMoW base alloy and the alloy welded with a dedicated CoCr wire after heat treatment.

Referring to the structure of the tested materials, the as-cast CoCrMoW base alloy showed a dendritic microstructure with inter-dendritic lamellar precipitates of secondary phases that became thicker and more visible after heat treatment. The use of filler materials during LBW resulted in structural refinement as compared to the non-welded one. Generally, signs of pit formation after electrochemical testing were preferentially located at the inter-dendritic regions.

Chemical compositions of the CoCr-welded joints showed enriched Mn and W depletion, which stemmed from the cobalt wire metal composition. The weld metal was diluted to ~50% with the base material shown in Figure 9a. The transient zone between CoCrMoW cast alloy and cobalt welding wire enriched Mo as well as Cr and Si from the cast alloy, which was concentrated predominantly within the inter-dendritic regions. The inter-dendritic regions were mostly uniform without complex lamellar structures. The limited carbon concentration in the inter-dendritic regions may preferentially lead to intermetallics rich in Co, Cr, and Si [12,19].

The heat treatment simulating PFM firing affected the homogenization of the transition zone, HAZ, and weld metal in terms of chemical composition toward a composition similar to the CoCrMoW base alloy. Heat treatment decreased Mo and increased W in the CoCr weld, while Mn levels dropped significantly. The CoCrMoW alloy welded with stainless steel wire was saturated in Fe and Mn, slightly increased in W, and showed dilution at ~30% relative to the base material (Figure 9b). Heat treatment affected interdiffusion between materials, which enriched Cr in the base CoCrMoW cast alloy, while the weld zone was enriched in Co and W. The transient zone between materials was enriched in Fe and depleted in Mn. This zone grew due to diffusion, showing a cellular microstructure with a strong segregation of alloying elements along coarsening cellular grain borders.

## 5. Conclusions

The studies of corrosion resistance in an artificial saliva environment of laser beam-welded (LBW) joints in various configurations of a CoCrMoW alloy with nickel-free stainless steel and cobalt alloy wire as well as autogenous welding, with subsequent heat treatment at 920–950 °C, simulating (PFM) firing, are summarized as follows:Welding of the CoCrMoW alloy for denture applications performed as autogenous welding (without filler material) and with the use of filler materials (stainless steel, CoCr wire) improved the corrosion resistance at steady-state potential conditions.The CoCrMoW alloy welded with filler materials showed lower corrosion properties than the CoCrMoW base alloy in non-welded conditions, as evidenced by the potentiodynamic results, in which the welded materials showed lower polarization resistances, or R_pol_.Comparing the corrosion resistance of CoCrMoW base alloy welded with two different wires (stainless steel and CoCr wire), slightly better electrochemical parameters were obtained with materials welded with stainless steel wire. Only the Bode diagram curve analyses showed greater homogeneity and stability when using CoCr wire as the filler metal.Comparing the polarization resistance R_pol_ from the Tafel analysis and the total charge transfer resistance Rp_(EIS)_ measured in EIS, the CoCrMoW alloy welded with a stainless steel wire after PFM heat treatment equaled or even slightly exceeded the corrosion resistance of the base alloy and alloy welded with dedicated CoCr wire after heat treatment.The structures of welded CoCrMoW alloys were dendritic and refined in the weld zone and transition zone (HAZ) due to the high LBW cooling rate, while the base material showed a coarse microstructure.The heat treatment simulating PFM firing affected the homogenization of the chemical composition in the transition between HAZ and the weld metal towards a composition of the CoCrMoW base alloy.It may be acceptable to use stainless steel wire instead of CoCr for welding CoCrMoW dental alloys using PFM. Such an interchangeable use of that filler material does not create a strong corrosive cell in artificial saliva.

## Figures and Tables

**Figure 1 materials-15-05721-f001:**
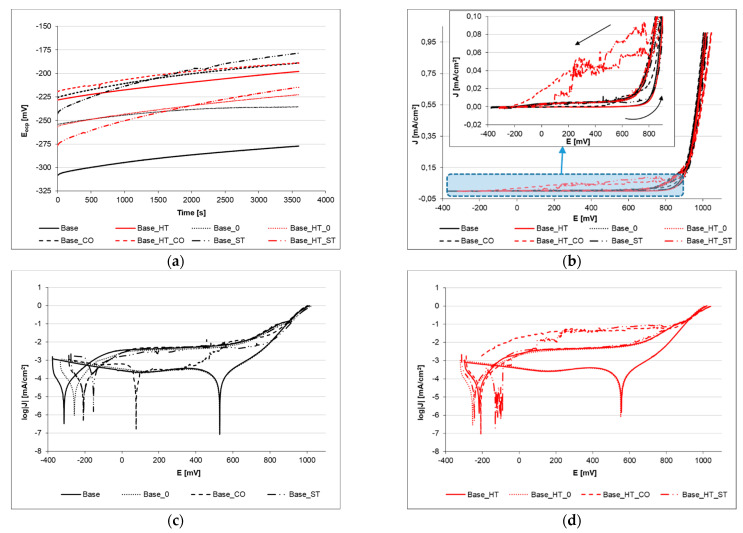
Electrochemical study results of welded CoCrMoW base alloy with (HT) and without (0) subsequent heat treatment for different filler materials—cobalt (CO) and stainless steel (ST) wires, as follows: (**a**) open circuit potential, (**b**) anodic polarization curves, (**c**,**d**) logarithmic presentation of anodic polarization curves for samples without and with HT.

**Figure 2 materials-15-05721-f002:**
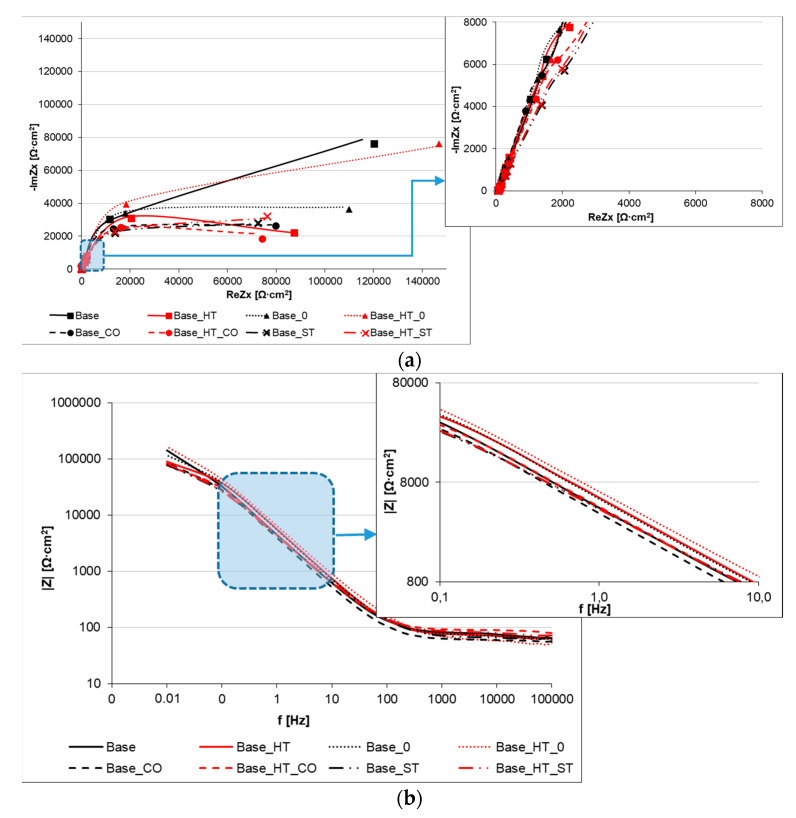
Impedance spectra of materials (for stationary potentials): (**a**) Nyquist representation; (**b**) Bode representation modulus vs. frequency, (**c**) Bode representation phase angle vs. frequency.

**Figure 3 materials-15-05721-f003:**
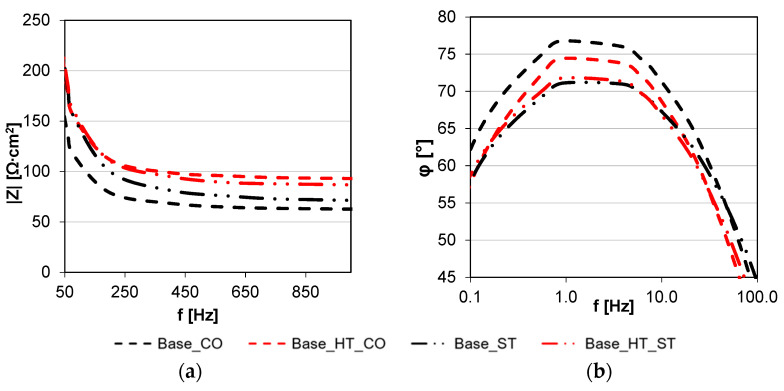
Bode representation of the impedance spectrum for materials welded with cobalt and stainless steel filler materials: (**a**) modulus vs. frequency, (**b**) phase angle vs. frequency.

**Figure 4 materials-15-05721-f004:**
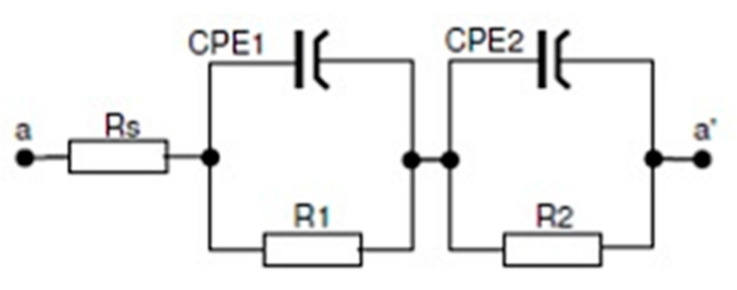
Equivalent electric circuits (EEC) for materials in artificial saliva solutions.

**Figure 5 materials-15-05721-f005:**
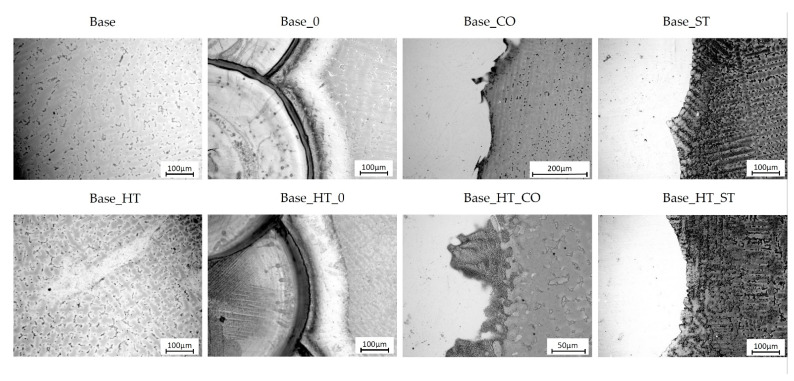
Microstructures of welded material configurations (LOM) after corrosion resistance tests, where base is non-welded CoCrMoW alloy, 0 is laser welding without filler metal, ST is welded with stainless steel wire, CO is welded with cobalt alloy wire, and HT refers to samples subjected to heat treatment after welding.

**Figure 6 materials-15-05721-f006:**
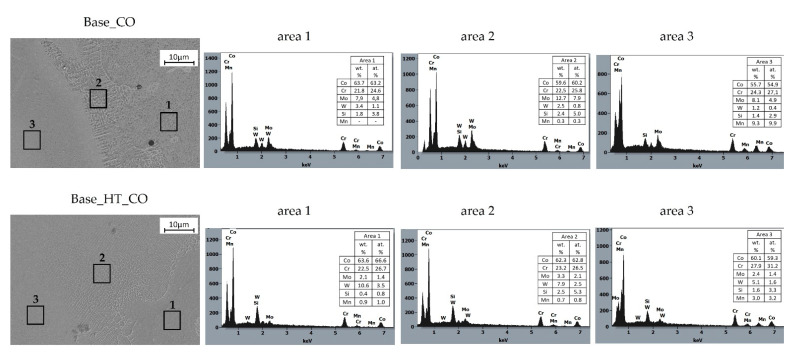
Microstructure and EDS analyses of CoCrMoW alloy welded with CoCr wire in as-welded conditions (Base_CO) and after heat treatment (Base_HT_CO).

**Figure 7 materials-15-05721-f007:**
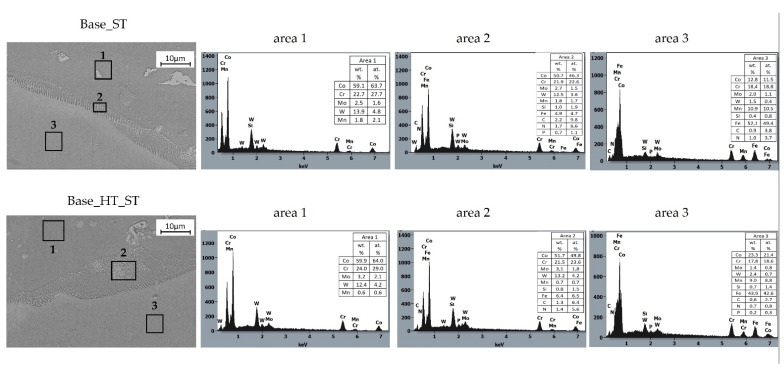
Microstructure and EDS analyses of CoCrMoW alloy welded with CoCr wire in as-welded conditions (Base_ST) and after heat treatment (Base_HT_ST).

**Figure 8 materials-15-05721-f008:**
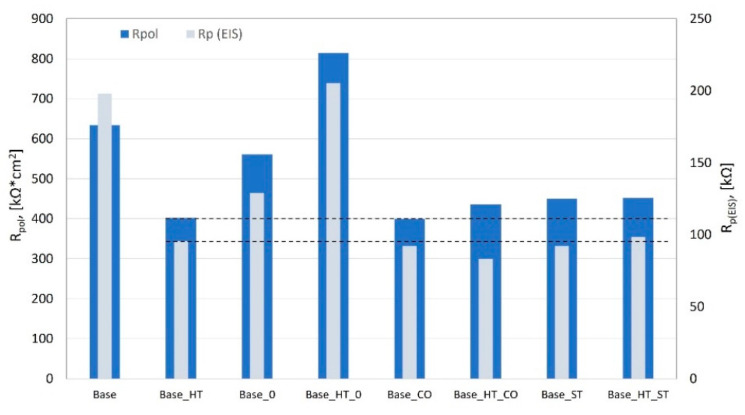
Comparison of polarization resistance values found by the Tafel and EIS methods.

**Figure 9 materials-15-05721-f009:**
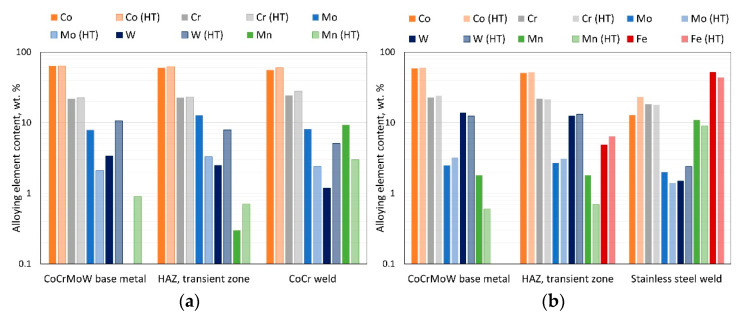
Comparison of the main alloying elements in the welded joints, as follows: (**a**) CoCrMoW welded with CoCr wire, (**b**) CoCrMoW welded with stainless steel wire.

**Table 1 materials-15-05721-t001:** Chemical compositions of the materials used.

Material (Commercial Name)	Element Concentration [wt. %]							
	Co	Cr	Mo	W	Si	Mn	N	Other
Base CoCrMoW alloy (Remanium 2000+)	61	25	7	5	1.5	<1	<1	-
Stainless steel wire (Menzanium 1.4456)	-	16–20	1.8–2.5	-	≤1	16–20	0.7–1.0	C ≤ 0.1; Ni ≤ 0.2; V ≤ 0.2; Fe rest
CoCr welding wire (Schütz Dental GmbH)	64.25	17.85	2.25	-	-	14.7	-	-

**Table 2 materials-15-05721-t002:** Conditions of the porcelain fused to metal (PFM).

Stage	Start Temperature [°C]	Heat Speed [°C/min]	Final Temperature [°C]	Time [min]
1	500	80	950	1
2		55	930	
3		55	920	

**Table 4 materials-15-05721-t004:** Potentiodynamic test results of welded joint configurations (base CoCrMoW alloy, autogenous welded base alloy, welding with CoCr wire, and stainless steel wire), with or without heat treatment.

Sample	E_ocp_	E_cor_	J_cor_	R_pol_	E_br_	E_rp_	r_cor_
	[mV]	[mV]	[nA/cm^2^]	[kΩ∙cm^2^]	[mV]	[mV]	[g/m^−2^∙year]
Base	−277	−313	21	634	900	908	0.30
Base_HT	−198	−217	41	402	916	961	0.58
Base_0	−236	−257	20	561	919	901	0.55
Base_HT_0	−222	−256	10	814	923	964	0.28
Base_CO	−189	−210	35	400	907	898	0.50
Base_HT_CO	−189	−210	21	436	934	909	0.29
Base_ST	−178	−207	12.7	450	929	968	0.18
Base_HT_ST	−215	−243	14.7	452	930	950	0.41

Abbreviations are as follows: Base—CoCrMoW alloy, HT—heat treatment, 0—laser beam autogenous welding (without filler metal), ST—welded with stainless steel wire, CO—welded with CoCr wire.

**Table 5 materials-15-05721-t005:** Impedance parameters of CoCrMoW alloy with and without heat treatment for two different filler materials.

Sample	R_s_	Y_1_	n_1_	R_1_	Y_2_	n_2_	R_2_
	[Ω]	[F∙s^(a−1)^]		[kΩ]	[F∙s^(a−1)^]		[kΩ]
Base	64	1.4 × 10^−6^	0.951	0.01	45.3 × 10^−6^	0.866	198
Base_HT	63	7 × 10^−3^	0.311	0.4	34.5 × 10^−6^	0.897	95
Base_0	62	3.1 × 10^−3^	0.372	2.0	35.7 × 10^−6^	0.903	127
Base_HT_0	51	1.4 × 10^−3^	0.456	1.0	32.3 × 10^−6^	0.878	204
Base_CO	57	1.5 × 10^−3^	0.557	2.0	50.8 × 10^−6^	0.913	90
Base_HT_CO	51	2.2 × 10^−3^	0.145	0.1	43.8 × 10^−6^	0.876	83
Base_ST	63	0.4 × 10^−3^	0.673	14	58.6 × 10^−6^	0.866	78
Base_HT_ST	71	8.2 × 10^−3^	0.276	0.4	49.7 × 10^−6^	0.847	98

## Data Availability

The study did not report any data.

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
