# Peer review of "Influence of Filler Metal on Electrochemical Characteristics of a Laser-Welded CoCrMoW Alloy Used in Prosthodontics"

_materials, 2022, doi:10.3390/ma15165721_

Round 1

Reviewer 1 Report

1.       Avoid forming unnecessarily lengthy run-on sentences in your manuscript. It can cause the reader to disconnect with the context easily and make your writing seem sloppy. It has been noticed more than twice just within the abstract. Please modify the sentences throughout the manuscript.

2.       There are several instances in the paper in which the authors have used inappropriate English sentence structure. It is highly recommended that the authors should revise the complete manuscript for correction of English language mistakes.

3.       What is the meaning of “40 ÷ 100 mV”, “920 ÷ 950°C” and “3 ÷ 5°”. I am confused about the usage of “÷ symbol. Same mistake has been repeated at many other instances in the manuscript.

4.       On page 4, line 167, change “testes” to “tested”.

5.       The * symbol is inappropriately used here in line 194, page # 5. “634 kΩ *cm2”. Same mistake has been repeated at other places in the manuscript including in legends of figure 2, 3 and 8. Please make all necessary corrections.

6.       Most of the references are not recent. It would be helpful to include most recent references in the related field.

7.       The scale bar in the SEM images is not clearly visible and is certainly not of the print quality. Please modify the figures with clear scale bars.

Author Response

First of all, we would like to thank you for your thorough reading of the article and valuable comments. The comments provided will contribute to the improvement of the article and our knowledge of the subject, all of them have been included in the manuscript. The article was checked by a native speaker and proofreading was done. The responses to the submitted comments of the reviewer are presented below.

  1. Avoid forming unnecessarily lengthy run-on sentences in your manuscript. It can cause the reader to disconnect with the context easily and make your writing seem sloppy. It has been noticed more than twice just within the abstract. Please modify the sentences throughout the manuscript.

The article has been revised linguistically. Too long sentences in the abstract and the entire manuscript have been shortened.

  1. There are several instances in the paper in which the authors have used inappropriate English sentence structure. It is highly recommended that the authors should revise the complete manuscript for correction of English language mistakes.

The article has been linguistically revised, the revised version was proofread.

  1. What is the meaning of “40 ÷ 100 mV”, “920 ÷ 950°C” and “3 ÷ 5°”. I am confused about the usage of “÷” Same mistake has been repeated at many other instances in the manuscript.

The symbol “÷” was changed to “-“ in all text.

  1. On page 4, line 167, change “testes” to “tested”.

Corrected as recommended.

  1. The * symbol is inappropriately used here in line 194, page # 5. “634 kΩ *cm2”. Same mistake has been repeated at other places in the manuscript including in legends of figure 2, 3 and 8. Please make all necessary corrections.

The symbol “*” was changed to “∙“ in all text, tables, and figures.

  1. Most of the references are not recent. It would be helpful to include most recent references in the related field.

Authors try to find and select only a few references in their opinion matched on a good level, some on them are really old but there are also some new and fresh.

years

number

years

number

years

number

years

number

years

number

1984-1999

3

2000-2005

4

2006-2010

8

2011-2015

5

2016-2021

4

The literature references were revised and updated.

  1. The scale bar in the SEM images is not clearly visible and is certainly not of the print quality. Please modify the figures with clear scale bars.

Corrected as recommended. The authors placed new figures 5, 6, and 7 with a larger scale bar.

Reviewer 2 Report

The author has done a lot of work to determine the corrosion resistance changes in artificial saliva of the CoCrMoW-based alloy used for dental prostheses. This work has a certain application prospect, but the work content needs to be further improved.

1 .“……the most popular joining method in dental engineering is the LBW……”How did you come to this conclusion?

2. The experimental design is relatively general. How to determine so many experimental conditions? It includes base alloy, heat treatment, laser welding without filler metal , welded with stainless steel wire, welded with CoCr alloy wire. Don't you consider the influencing factors of single conditions?

3. What is the equipment used for electrochemical performance test? Whether the test process error is considered?

4.Why is the change so large at the initial stage of the open circuit potential curve as shown in FIG.1? Not just this curve of Base_ST. The author should analyze the reasons in depth. In addition, why use two pictures (FIG.1b and FIG. 1c) to illustrate the same experiment?

5.For the potentiodynamic polarization process of this material CoCrMoW. What kind of reaction happens in different polarization stages? The author should give analysis or discussion.

6.In order to describe microstructural characterizations more accurately, the author should unify the scale as shown in FIG.5.

7.This conclusion needs to be improved, which contains a lot of speculation or meaningless content.

Author Response

First of all, we would like to thank you for your thorough reading of the article and valuable comments. The comments provided will contribute to the improvement of the article and our knowledge of the subject, all of them have been included in the manuscript. The article was checked by a native speaker and proofreading was done. The responses to the submitted comments of the reviewer are presented below.

The author has done a lot of work to determine the corrosion resistance changes in artificial saliva of the CoCrMoW-based alloy used for dental prostheses. This work has a certain application prospect, but the work content needs to be further improved.

1 .“……the most popular joining method in dental engineering is the LBW……”How did you come to this conclusion?

The sentence was corrected

“Presently, the most promising joining methods in dental engineering is the LBW because it does not require a unique filler material; this prevents the possibility of a galvanic cell between materials and the consequent reduction in corrosion resistance [3].”

  1. The experimental design is relatively general. How to determine so many experimental conditions? It includes base alloy, heat treatment, laser welding without filler metal , welded with stainless steel wire, welded with CoCr alloy wire. Don't you consider the influencing factors of single conditions?

The presented research is the result of previous work in this area. Optimized welding conditions of the alloys tested, and the applied heat treatment are selected. The aim of the work was to compare the use of various binders for welding CoCrMoW alloy, and the binders themselves were selected on the basis of those available on the market and used in dental practice. For this reason, comparative tests of corrosion resistance for constant welding conditions were performed and variable factors for individual conditions were not analyzed.

  1. What is the equipment used for electrochemical performance test? Whether the test process error is considered?

Equipment used for electrochemical performance vas revised and provided in the paper, in the methodology part: Corrosion tests were performed on the potentiostat station Atlas 0531EU&IA (At-las-Sollich, Poland) (…). The electrochemical tests were carried out in three-electrode corrosion cell systems, (…) with a (…) reference electrode of Ag/AgCl (potential 207 mV at 25°C), and a platinum wire as the auxiliary electrode.

The electrochemical tests were performed for two samples for each state and because the results were very close, they decided to skip the test for the next samples. One representative result was selected. This information was added to the article.

4.Why is the change so large at the initial stage of the open circuit potential curve as shown in FIG.1? Not just this curve of Base_ST. The author should analyze the reasons in depth. In addition, why use two pictures (FIG.1b and FIG. 1c) to illustrate the same experiment?

Information about the upward trend of free potential was added to the article. It will be worth paying attention to it in future studies and perhaps, for research purposes, to extend the registration time.

The registered upward trend, observed on Figure 1a, in the study of the stationary potential may be related to the appearance of its ions at the metal-electrolyte interface in order to create an equilibrium state.

The authors presented the results of the potentiodynamic curves using two pictures because in Figure 1b it is much easier to observe a clear inflection related to the breakdown potential, while Figure 1c is a characteristic fragment for Tafel's analysis.

5.For the potentiodynamic polarization process of this material CoCrMoW. What kind of reaction happens in different polarization stages? The author should give analysis or discussion.

Information about polarization stages was added into text.

The tested materials were polarized at a rate of 1 mV/s until a current density of 1 mA/cm2 was reached, and then the direction of polarization was reversed. The determined characteristic potentials allowed to distinguish in Figures 1c and 1d the passivity zone, visible as a section of the horizontal line after the corrosion potential peak. For potentials higher than Ebr, the pitting initiation can be observed, while for potentials more negative than Erp, the existing pits were repassivated. A path in the range of the Ebr to Epr potentials was safe with regard to new pits, but those that exist on the surface could develop.

6.In order to describe microstructural characterizations more accurately, the author should unify the scale as shown in FIG.5.

Corrected as recommended. The figure 5, 6 and 7 were corrected, and scale bars were modified.

7.This conclusion needs to be improved, which contains a lot of speculation or meaningless content.

The conclusions were corrected and improved.

Reviewer 3 Report

Nice work and it can be published after considering following remarks:

1- On page 2 line 52 what kind of discontinuity is significant? a void??

2- The criteria for selecting filler material is missing, on what basis filler materials were selected.

3- Pg 3, line 105, pls give reference to the relevant procedure for manufacturing dentures.

4- Pg 3, Line 115, "270÷290 V" an unusual way of representing voltage values, pls correct it. similarly for (83÷90 kΩ) in Pg 9, Line 294. Please correct them all.

5- Pls provide a reference for the Fusayamya formula in Table 3.

6- Pls use proper reference to the Fig and Tables i.e. see Table 1 or refer to Table 1, do not write (tab 1).

7- Pls give justification for the use of processing parameters mentioned in Table 2.

8- I would suggest redrawing Fig 1c by making separate Figs for (HT) and without heat treatment.

9- Pls use write units properly e.g. in kΩ *cm2, "*" can be replaced with a dot. Similarly for g/m-2*year-1.

10- In EIS Fig 2a may be replotted to show both the fitting lines and experimental data.

11- How many tests were performed in establishing EIS and PD results? what was the Data repeatability?

Author Response

First of all, we would like to thank you for your thorough reading of the article and valuable comments. The comments provided will contribute to the improvement of the article and our knowledge of the subject, all of them have been included in the manuscript. The article was checked by a native speaker and proofreading was done. The responses to the submitted comments of the reviewer are presented below.

1- On page 2 line 52 what kind of discontinuity is significant? a void??

The authors mean a discontinuity resulting, for example, from the preparation of two parts of one prosthesis to be joined, which may be 0.5 even to 1.0 mm.

2- The criteria for selecting filler material is missing, on what basis filler materials were selected.

The selection of filler materials was made according to the criterion of their chemical composition as closely as possible to the chemical composition of the welded alloy. Therefore, one of the materials is chromium-cobalt wire with molybdenum addition, and the other is steel with a chromium content similar to that of the base alloy.

An explanatory sentence was added to the article.

3- Pg 3, line 105, pls give reference to the relevant procedure for manufacturing dentures.

A reference containing the procedure for manufacturing fixed dentures has been added:

[14] Spiechowicz E.; Dental prosthetics, Medical Publishing House PZWL, Warsaw 2008

4- Pg 3, Line 115, "270÷290 V" an unusual way of representing voltage values, pls correct it. similarly for (83÷90 kΩ) in Pg 9, Line 294. Please correct them all.

The notation was corrected.

5- Pls provide a reference for the Fusayamya formula in Table 3.

Done. Authors specified the name Fusayama/Meyer artificial saliva and added a citation.

6- Pls use proper reference to the Fig and Tables i.e. see Table 1 or refer to Table 1, do not write (tab 1).

The notation was corrected.

7- Pls give justification for the use of processing parameters mentioned in Table 2.

The heat treatment conditions presented in Table 2 are to simulating the porcelain firing on metal process for ceramic material VMK Master from Vita company. Information about source of heat treatment conditions was added to article.

8- I would suggest redrawing Fig 1c by making separate Figs for (HT) and without heat treatment.

The authors divided Figure 1c into two separate drawings, one for samples without HT and the second with HT.

9- Pls use write units properly e.g. in kΩ *cm2, "*" can be replaced with a dot. Similarly for g/m-2*year-1.

The symbol “*” was changed to “∙“ in all text, tables, and figures.

10- In EIS Fig 2a may be replotted to show both the fitting lines and experimental data.

The figure was replotted. The authors showed in Figure 2a experimental data by chart markers and fittings lines.

11- How many tests were performed in establishing EIS and PD results? what was the Data repeatability?

The electrochemical tests were performed for two samples for each state and because the results were very close, they decided to skip the test for the next samples. One representative result was selected.

This information was added to the article.

Reviewer 4 Report

The authors present results related to the influence of the filler metal on electrochemical characteristic of ally used in prosthodonics. I found article confusing written whit a failure of interpreting electrochemical results. The interpreting does not fit to the experimental results, as well as experimental results does not match each other.

Abstract:

line 18:

"Welding CoCrMoW alloy without and with a filler material increased the open     

circuit potential of the samples by 40 ÷ 100 mV".

COMMENT: This sentence is confusing. It seems that there is no difference if welding was done including filler material. Please clarify it.

line 25-31:

COMMENT: The sentence that described EIS results must be written in "spirit of EIS". I found inappropriate to describe an "angles" in Nyquist diagram in Abstract. The authors need to offer definite conclusion based on relevant experimental data. Also "….higher impedance modulus…in the entire range of studied frequency…" can not be argument for given conclusion, as different processes take place at different frequency. Please rely your conclusion on the processes that describe charge transfer process.

Materials and methods:

line 120: ground paper size and polishing method must be given.

Table 1. :  Dissolution of mentioned salts and urea in 1 L of water does not give concentration that response to Fusayamya formula.

line 147: What means "…forced flow (of current) through the solidified AC system…"?

line 148: Frequency range does not respond to the experimental results presented in Figure 2.  By applying mentioned frequency range, only ohmic resistance of solution can be determined and other very, very fast processes.

Results:

Figure 1: Figure 1 contains a too many dependences, and it is difficult to follow the Manuscript together with presented results. In addition, nothing important can not be seen from Figure 1b, since a large value of the current density mask a important part of graph.

Also, methods for calculation (or determination) of Rpol, Ebr, Erp must be given!

In Figure 1c, there is tremendous number of curves. It seems there is double curve for each material. For example, I found two corrosion currents and potential for each material (for example for black curve-Base, there is one at around -0.3 V and second at +0.55 V). What this means. Authors mist give procedure in detail for recording anodic polarization curves. Thius is also confusing, since from Figure 1b (as I can conclude) there is no transition between cathodic and anodic current at potential lower than -0.2 V. However, I am not sure since presentation in Figure 1b is poor and confusing. 

Approach to the explanation of the Spectra of the Electrochemical Impedance Spectroscopy measurement is also poor and confusing. 

The authors use terms and explanation which is strange to the readers who are expert in EIS.

line 222: "…that in a wide range of tested frequencies, significantly lower susceptibility to    

corrosion…"

COMMENT: What is dependence of the corrosion rate vs. frequency?

line 223-227: "This can be deduced from the high impedance value (imaginary part of impedance ImZx) at the variable frequency for those samples. Identical results were recorded for high-frequency values  (left site in X-axis on the Nyquist plot, presented in magnification), showing the highest  slope of the curve, which corresponds to the lowest corrosion rate."

COMMENT: Nyquist diagram in fact represent phase angle! What can be conluded from phase angle? What is parameter that affect phase angle? What is the approach that allow conclusion related to corrosion resistance at HF?

From the Nyquist diagram I found only solution resistance and capacitive behaviour at HF and MF!

Also, Bode diagram and Nyquist diagram does not match! Also, at low frequency from the Figure 2 c it can be concluded that phase angle increases while in Figure 1a it can be seen that decreases whit decrease of f.

line 232: frequency range is confusing (from 100 kHZ to 660 MHz?!)….

The authors must comment inductive behaviour that can be sees from Figure 2c at HF!

Inset of Figure 2b is not clear? What it represent and how it is relevant for corrosion.

Figure 4: The used EEC does not fit to experimental data! I made simulation of data presented from Table 5, although it can be concluded intuitively!

line 272-274:"… CPE 1  element models the capacity of the surface zone of the material, the R2 is the charge transfer resistance across the interface, and CPE 2  can    

be thought as reflecting the electrical double layer at the material-solution interface."

COMMENT: Please consult relevant literature for values of double-layer capacity. The authors must consult relevant literature and books concerning EIS and interpreting EIS data.

line 280-281: What a meaning?! There is values for "n" and suddenly "a" is given!? The Warburg element is used or not (there is a=1 for Base material?!)!?

What is difference between R1 and R2? From the inrellevant data I can conclude that R1 represent polarization resistance!

Author Response

First of all, we would like to thank you for your thorough reading of the article and valuable comments. The comments provided will contribute to the improvement of the article and our knowledge of the subject, all of them have been included in the manuscript. The article was checked by a native speaker and proofreading was done. The responses to the submitted comments of the reviewer are presented below.

The authors present results related to the influence of the filler metal on electrochemical characteristic of ally used in prosthodonics. I found article confusing written whit a failure of interpreting electrochemical results. The interpreting does not fit to the experimental results, as well as experimental results does not match each other.

Abstract:

line 18:

"Welding CoCrMoW alloy without and with a filler material increased the open circuit potential of the samples by 40 ÷ 100 mV".

COMMENT: This sentence is confusing. It seems that there is no difference if welding was done including filler material. Please clarify it.

The sentence was corrected.

Welding CoCrMoW alloy without and with a filler material increased the open circuit potential of the samples by 40 - 100 mV compared to unwelded the base alloy.

The abstract was rewritten.

line 25-31:

COMMENT: The sentence that described EIS results must be written in "spirit of EIS". I found inappropriate to describe an "angles" in Nyquist diagram in Abstract. The authors need to offer definite conclusion based on relevant experimental data. Also "….higher impedance modulus…in the entire range of studied frequency…" can not be argument for given conclusion, as different processes take place at different frequency. Please rely your conclusion on the processes that describe charge transfer process.

The abstract was rewritten.

Materials and methods:

line 120: ground paper size and polishing method must be given.

Authors add information about the preparation of samples: ground with SiC paper, grade 500, 800, and 1200, and then polished with a diamond suspension of 3 µm.

Table 1. :  Dissolution of mentioned salts and urea in 1 L of water does not give concentration that response to Fusayamya formula.

The authors specified the name of solution as Fusayama/Meyer artificial saliva, a similar formula of Fusayama/Meyer solution can be found in other works:

Kinani L., Chtaini A.; Corrosion Inhibition of Titanium in Artificial Saliva Containing Fluoride, Leonardo Journal of Sciences 11 (2007) 33-40.

Matykina E., Arrabal R., Mohedano M. et al.; Stability of plasma electrolytic oxidation coating on titanium in artificial saliva. J Mater Sci: Mater Med 24 (2013) 37-51; https://doi.org/10.1007/s10856-012-4787-z

Reimann, Ł.; Electrochemical Characteristics of a Cobalt Alloy with a Protective Passive Layer, Archives of Metallurgy and Materials 61/3 (2016) 937-944

line 147: What means "…forced flow (of current) through the solidified AC system…"?

The authors correct this sentence to: (…) 15 min without current flow and then with a flow of current through the circuit at an amplitude of 10 mV in the frequency range from 100 kHz to 10 mHz.

line 148: Frequency range does not respond to the experimental results presented in Figure 2.  By applying mentioned frequency range, only ohmic resistance of solution can be determined and other very, very fast processes.

The authors corrected the frequency range from 100 kHz to 10 mHz, and in other places of work.

Results:

Figure 1: Figure 1 contains a too many dependences, and it is difficult to follow the Manuscript together with presented results. In addition, nothing important can not be seen from Figure 1b, since a large value of the current density mask a important part of graph.

Also, methods for calculation (or determination) of Rpol, Ebr, Erp must be given!

In Figure 1c, there is tremendous number of curves. It seems there is double curve for each material. For example, I found two corrosion currents and potential for each material (for example for black curve-Base, there is one at around -0.3 V and second at +0.55 V). What this means. Authors mist give procedure in detail for recording anodic polarization curves. Thius is also confusing, since from Figure 1b (as I can conclude) there is no transition between cathodic and anodic current at potential lower than -0.2 V. However, I am not sure since presentation in Figure 1b is poor and confusing. 

Authors presented a close-up of the indicated range on the Figure 1b and divided Figure 1c on two separate graphs, one for samples without HT and second with HT. The potentiodynamic research were presented as whole hysteresis look, because after current density reaching 1 mA/cm2 the polarity was reversed, and potential decreasing to the initial value.

In parts in the methodology information on the method of calculation of the polarization resistance according to the Stern-Geary equation (1) and the method of determination of the breakdown potential (Ebr) as a place of depassivation and inflection of the anode curve as well as the method of determination of the value of repassivation potential (Erp) as the point of intersection of the return and primary curve has been added.

Approach to the explanation of the Spectra of the Electrochemical Impedance Spectroscopy measurement is also poor and confusing. 

The authors use terms and explanation which is strange to the readers who are expert in EIS.

line 222: "…that in a wide range of tested frequencies, significantly lower susceptibility to corrosion…"

COMMENT: What is dependence of the corrosion rate vs. frequency?

The sentence was corrected.

line 223-227: "This can be deduced from the high impedance value (imaginary part of impedance ImZx) at the variable frequency for those samples. Identical results were recorded for high-frequency values  (left site in X-axis on the Nyquist plot, presented in magnification), showing the highest  slope of the curve, which corresponds to the lowest corrosion rate."

COMMENT: Nyquist diagram in fact represent phase angle! What can be conluded from phase angle? What is parameter that affect phase angle? What is the approach that allow conclusion related to corrosion resistance at HF?

From the Nyquist diagram I found only solution resistance and capacitive behaviour at HF and MF!

The paper was updated with this content.

The value of the phase shift angle, i.e. the delay of the system's response to the voltage signal, changes the impedance value. In the case where the current and voltage are in phase the impedance can be expressed as pure resistance, when φ takes values close to -90° the impedance can be described by a capacitor, and for φ going to + 90° like a inductor. The course of the phase shift angle curves and its value allow the assessment of the quality of the protective barrier against corrosion damage on the surface of the tested material. The smallest value of the phase shift angle allows to assume that this barrier is the thickest and compact.

Also, Bode diagram and Nyquist diagram does not match! Also, at low frequency from the Figure 2c it can be concluded that phase angle increases while in Figure 1a it can be seen that decreases whit decrease of f.

Figure 2b and 2c were corrected because it presented other range of frequencies that mentioned in methodology: from 100 kHz do 10 mHz.

line 232: frequency range is confusing (from 100 kHZ to 660 MHz?!)….

The frequency range was corrected: from 100 kHz do 660 mHz.

The authors must comment inductive behaviour that can be sees from Figure 2c at HF!

Authors added to article in section Results information about the inductive behavior:

In Figure 2c, in the high frequency range of 40-100 kHz, a decrease in the value of the phase angle was observed, corresponding to the assumption of positive values by the impedance component ImZx. This phenomenon is probably related to the fact that in high frequencies the impedance of the parasitic capacitor decreases while the impedance of the inductor increases and modeled element behaves like a inductor.

Inset of Figure 2b is not clear? What it represent and how it is relevant for corrosion.

Authors have posted Figure 2b in order to more fully present the frequency characteristics using the Bode plot, and for better visualization they have separately presented the amplitude (Figure 2b) and phase (Figure 2c) relationships. The presentation of changes in the impedance modulus of the corrosion systems allows to suppose that its higher value is related to the existence of a better barrier on the surface for the formation of corrosion damage.

The paper was updated with this content.

Figure 4: The used EEC does not fit to experimental data! I made simulation of data presented from Table 5, although it can be concluded intuitively!

Authors to fit the experimental data to the electrical circuit used EC-Lab software (Bio-Logic Science Instrument) and presented the best results from it, in addition, in Figure 2a with the impedance spectra, authors added markers of the values determined in the experiment and the extrapolation line. By the way, Table 5 was corrected by changing the sequence of R1 and CPE1 values with R2 and CPE2.

line 272-274:"… CPE 1  element models the capacity of the surface zone of the material, the R2 is the charge transfer resistance across the interface, and CPE 2 can be thought as reflecting the electrical double layer at the material-solution interface."

COMMENT: Please consult relevant literature for values of double-layer capacity. The authors must consult relevant literature and books concerning EIS and interpreting EIS data.

The paper was updated with this content. The explanation of R1 and R2 resistances was provided and linked their physical meaning.

line 280-281: What a meaning?! There is values for "n" and suddenly "a" is given!? The Warburg element is used or not (there is a=1 for Base material?!)!?

Authors corrected the designation of the coefficient of CPE element as “n” in whole text, and correct the value of n for Base sample.

What is difference between R1 and R2? From the inrellevant data I can conclude that R1 represent polarization resistance!

The paper was updated with this content. The explanation of R1 and R2 resistances was provided and linked their physical meaning.

Round 2

Reviewer 2 Report

The revised manuscript is acceptable.

Reviewer 4 Report

My first decision was "reject".   I have nothing against that the manuscript is in revision process, however I found inappropriate to send revised version of the manuscript again to me. After first revision round, i found my job finished.